# Early Lesion of Post-Primary Tuberculosis: Subclinical Driver of Disease and Target for Vaccines and Host-Directed Therapies

**DOI:** 10.3390/pathogens10121572

**Published:** 2021-12-02

**Authors:** Robert E. Brown, Robert L. Hunter

**Affiliations:** Department of Pathology and Laboratory Medicine, University of Texas Health Sciences Center at Houston, Houston, TX 77030, USA; robert.brown@uth.tmc.edu

**Keywords:** tuberculosis, secretory antigens, bronchiolar epithelium, alveolar pneumocytes, M2 polarization, COX-2, FAS, pathogenesis, early lesion

## Abstract

The characteristic lesion of primary tuberculosis is the granuloma as is widely studied in human tissues and animal models. Post-primary tuberculosis is different. It develops only in human lungs and begins as a prolonged subclinical obstructive lobular pneumonia that slowly accumulates mycobacterial antigens and host lipids in alveolar macrophages with nearby highly sensitized T cells. After several months, the lesions undergo necrosis to produce a mass of caseous pneumonia large enough to fragment and be coughed out to produce a cavity or be retained as the focus of a post-primary granuloma. Bacteria grow massively on the cavity wall where they can be coughed out to infect new people. Here we extend these findings with the demonstration of secreted mycobacterial antigens, but not acid fast bacilli (AFB) of *M. tuberculosis* in the cytoplasm of ciliated bronchiolar epithelium and alveolar pneumocytes in association with elements of the programmed death ligand 1 (PD-L1), cyclo-oxygenase (COX)-2, and fatty acid synthase (FAS) pathways in the early lesion. This suggests that M. tuberculosis uses its secreted antigens to coordinate prolonged subclinical development of the early lesions in preparation for a necrotizing reaction sufficient to produce a cavity, post-primary granulomas, and fibrocaseous disease.

## 1. Introduction

Many investigators today believe that cavities form by erosion of caseating granulomas into bronchi. Pathologists in the pre-antibiotic era knew that this is wrong. Cavities form by dissolution of caseous pneumonia, not by erosion of granulomas [1]. Earlier investigators conducted many autopsies and so were familiar with a whole series of tuberculous lesions that are seldom seen today. They recognized that the onset of clinical post-primary tuberculosis (PPTB) is preceded by 1–2 years of asymptomatic subclinical development of the early lesion in the lung before appearance of caseous pneumonia that is either coughed out to form cavities or is retained to become the focus of post-primary granulomas and fibrocaseous disease [2,3,4]. Organisms then grow massively on the surface of cavities where they can be coughed into the environment. Today, since there is no medical reason to biopsy the early lesions and autopsies are infrequent, most investigators are unaware of their existence even though they remain familiar to radiologists as the “tree-in-bud” sign characteristic of developing PPTB [5].

The early lesions of PPTB do not begin until after establishment of sufficient immunity to prevent disease in all extra pulmonary and most pulmonary sites [1]. PPTB begins as an alveolitis that spreads for months via bronchi as an obstructive lobular pneumonia recognizable on high resolution CT scans as the “tree-in-bud” sign [5]. The early lesions are paucibacillary and consist of alveolar macrophages that slowly accumulate secreted mycobacterial antigens and host lipids. They become foamy and accumulate behind obstructed bronchi in close association with highly sensitized T cells [6,7]. Many lesions regress spontaneously, but some undergo necrosis to produce caseous pneumonia that is either coughed out to form cavities or is retained to become the focus of post-primary granulomas and fibrocaseous disease [6]. Once a cavity is formed, masses of bacteria proliferate on its surface where they can be coughed out into the environment to infect new hosts.

Post-primary granulomas are easily identified because they form to surround preexisting foci of caseous pneumonia and therefore contain ghosts of alveoli rather than the homogeneous caseum of primary granulomas [1]. Primary and post-primary granulomas are seldom found together in the same lung. We reported sequestration of secreted mycobacterial antigens in foamy alveolar macrophages in the developing lesions of PPTB and that they are released with the onset of caseation necrosis in a fashion suggestive of the Koch phenomenon [6].

Here we report finding M. tuberculosis (MTB) antigens, but not acid-fast bacilli (AFB), in alveolar lining cells (pneumocytes) and ciliated bronchial cells together with elements of the programmed death ligand 1 (PD-L1), cyclo-oxygenase (COX)-2, and fatty acid synthase (FAS) pathways in developing lesions of PPTB. This suggests that MTB use secreted antigens to orchestrate an extended sequence of subclinical lesions that prepare for a sudden massive necrotizing pneumonia sufficient to produce cavities from which the organisms can escape to new hosts. Post-primary granulomas arise to surround foci of caseous pneumonia that are not coughed out to form cavities. They persist to become fibrocaseous disease. Most nascent post-primary lesions regress spontaneously leaving apical scars. If we understood why they regress, it might be possible to make them all regress and thereby drive MTB to extinction.

## 2. Results

Histologic slides from over 50 autopsies of people who died of untreated TB were examined to identify tissue sections with the early lesion of PPTB as previously described [1,7]. Briefly, the early lesions are an obstructive lobular pneumonia that can be distinguished from non-specific inflammation by the presence of mycobacterial antigens in lipid-rich foamy alveolar macrophages. This disease occurs preferentially in immunocompetent young adults with strong tuberculin skin tests [2,3,4]. Since all of the patients in this study were adults who died of pulmonary TB, they all had advanced post-primary lesions with little or no extra pulmonary TB. However, since the lesions of PPTB develop independently of one another, one commonly finds early lesions in the same lung as advanced lesions. Once the appropriate sections and fields were identified, the results were consistent among all cases.

The early lesion of PPTB is a post-obstructive lobular lipid pneumonia that spreads via bronchi for months prior to caseation [1,6,8] (Figure 1). Secreted MTB antigens in the early post-primary phase were detected by immunohistochemistry within alveolar macrophages (Figure 1A), alveolar pneumocytes (Figure 1B), and ciliated bronchiolar epithelial cells (Figure 1C). Alveolar monocytes and macrophages are not necessarily foamy in the early post-primary phase and show only mild expression of MTB antigen compared with the later phases. The mycobacterial antigens were diffusely present throughout the cytoplasm of the cells as is characteristic of secreted antigens rather than the discrete spots characteristic of intact organisms [8,9]. PD-L1 expression was present on alveolar pneumocytes, sloughed bronchiolar epithelial cells, alveolar monocytes, and macrophages (Figure 2A). PD-1 staining was present on lymphocytes primarily in the interstitium (Figure 2B). COX-2 expression was evident by immunohistochemistry within reactive alveolar pneumocytes (Figure 2C) and bronchiolar epithelial cells (Figure 2D). Alveolar macrophages were evident in the early lesion of PPTB stain with CD163, a marker for M2 polarized macrophages (Figure 2E). Finally, FAS expression was present in the cytoplasmic compartment of reactive alveolar pneumocytes surrounding increasingly foamy alveolar macrophages (Figure 2F). None of these markers were present to any appreciable extent outside of the TB lesions (Figure 2G), and the controls were negative for all markers within the lesions (Figure 2H).

## 3. Discussion

Since MTB is an obligate human parasite, everything that it does has been selected to ensure survival and transmission among people. MTB secretes many protein antigens that are required for virulence in humans but are not necessary for growth in culture [10]. The functions of some secreted proteins such as ESAT-6 and Ag85 have been studied, but most remain unknown [9,10]. Our finding of secreted mycobacterial antigens in multiple cell types of the early lesion of PPTB suggests a new synthesis of its pathogenesis.

The early lesion of PPTB is an asymptomatic obstructive lobular pneumonia that develops for 1–2 years before onset of symptoms [11]. It has been observed and described by multiple investigators for over a century but has never been reported in any animal model [2,3,4,12]. MTB is an obligate human parasite because no animal produces the PPTB lesions that mediate transmission to new hosts. The early lesion is a prolonged subclinical accumulation of mycobacterial antigens and host lipids in alveolar macrophages, near highly sensitized T cells with little inflammation, even though there may be intense tuberculous inflammation elsewhere in the same lung [8]. The process slowly spreads via bronchi in a process known as bronchogenic TB [5,13].

The finding of MTB antigens in multiple types of cells of the early lesion suggests that MTB use these antigens to coordinate diverse components of the host response to subclinically produce the materials necessary for a necrotizing reaction sufficient to form a cavity large enough to support transmission of infection. The components of the early lesion include:Bronchial obstruction traps alveolar macrophages to produce post-obstructive lipid pneumonia [14].Alveolar lining cells use FAS to produce lipids.Alveolar macrophages with M2 phenotype (CD163 Staining) become foamy by accumulating host lipids and secreted mycobacterial antigens.Sensitized tissue resident T cells (TRM) accumulate in alveolar walls.PD-L1 expression on alveolar macrophages and alveolar pneumocytes suppress PD-1^+^ T cell activity.

These lesions spread via bronchi as an obstructive lobular pneumonia, recognized by CT scans as the “tree-in-bud” sign, until they either regress or undergo necrosis to become caseous pneumonia that is either coughed out to form cavities or retained to become the focus of post-primary granulomas [1].

Our findings include (1) demonstration of mycobacterial antigen within both the type 1 and type 2 alveolar pneumocytes and ciliated bronchiolar lining cells in addition to alveolar monocytes/macrophages; (2) COX-2 expression in reactive alveolar pneumocytes and ciliated bronchiolar epithelium; (3) expression of FAS that could contribute fatty acids to the foamy alveolar macrophages; (4) expression of programmed death ligand (PD-L1) on the alveolar monocytes/macrophages, pneumocytes, and bronchiolar epithelial cells; and (5) CD163 expression on monocytes/macrophages in alveoli.

In previous studies, we reported that the early lesion of PPTB contains foamy alveolar macrophages with the M2 phenotype that marked with CD163, PD-L1, phosphorylated mTOR, insulin-like growth factor-1 receptor (IGF-1R), and human cyclooxygenase 2 (COX-2) [15]. Relatively few CD4^+^ cells were present, but abundant CD8^+^, PD-1^+^ cells were in the alveolar walls. Foamy macrophages stained strongly with CD68 and frequently also with the dendritic cell marker DEC-205. T-regulatory cells were also observed in the early lesion [16,17]. These findings suggest that MTB creates a protective microenvironment that accumulates high concentrations of MTB antigens and sensitized T cells that when released lead to necrosis, cavitation, post-primary granulomas, and fibrocaseous disease [17].

The early lesion of PPTB appears to be an attractive target for host directed therapies [15,18]. Mice, guinea pigs, and rabbits all develop infections that at certain points appear to be models of stages of human PPTB [19]. Evidence for a mechanism of host-directed therapy is provided by studies of FAS. The presence of FAS in alveolar lining cells of the early lesion of PPTB indicates that increased synthesis of lipids is a critical component of these lesions. We demonstrated that inclusion of a FAS inhibitor, lactoferrin, with a BCG vaccine in mice induced a sustained reduction in lung pathology but not numbers of organisms in tissue [20]. Metformin and orlistat are also inhibitors of FAS [21,22,23,24] and have shown efficacy against TB in preclinical studies [20,22]. Multiple studies of vitamin D3, metformin, and indomethacin on human specimens in vitro also suggest efficacy against the early lesion. In two separate human trials, metformin treatment was associated with improved control of infection and decreased disease severity [24,25]. Vitamin D3 has also been shown to reduce the accumulation of lipids by macrophages and to produce beneficial effects in patients in a controlled trial of pulmonary tuberculosis [26,27].

Several lines of evidence also suggest that the early lesion might also be an attractive target for vaccines. First, many nascent lesions regress spontaneously. If we knew why, it might be possible to induce all to regress. Second, slowly progressive pulmonary tuberculosis in the mouse is a model of the early lesion of PPTB [28]. Several experimental vaccines have been shown to prevent progression of these lesions [19,20,29]. Finally, the M72/AS01E vaccine that was successful in a human clinical trial was designed to induce a strong Th1 macrophage response without a M2 macrophage response [30,31]. We propose that the protection was provided by suppressing the M2 response and thereby preventing development of the early lesion of PPTB.

## 4. Material and Methods

Formalin-fixed and paraffin-embedded tissue blocks of human tuberculous lung samples were obtained during regular autopsy practice and after the completion of all medical, legal, and ethical requirements and were deidentified. Information provided included the age, sex, pulmonary TB as the primary cause of death, and negative HIV. Drug sensitivities were not known. Furthermore, this study was conducted according to the principles expressed in the Declaration of Helsinki. Hematoxylin and eosin (H&E) staining was used to identify blocks with the early lesion of PPTB [1]. Grossly, the early lesion of PPTB may be only a mild thickening of lung tissue. Consequently, it is necessary to prepare multiple sections of lung tissue for microscopic and immunohistochemical examination in order to identify them.

Immunohistochemistry was performed on five micrometer sections that were deparaffinized and stained with monoclonal antibodies conjugated with 3,3′-diaminobenzidine (DAB) for detection by immunohistochemistry. The antibodies used were MTB antigen (ab905, Abcam, Cambridge, MA, USA, cyclo-oxygenase (COX)-2 (SP21rabbit, Biocare Medical, Pacheco, CA, USA), programmed death-ligand 1 (PD-L1: Spring Bioscience, Pleasanton, CA, USA), programmed death-1 (PD-1), (Biocare Medical, Concord, CA, USA), and fatty acid synthase (FAS:C20G5, Cell Signaling Technology, Danvers, MA, USA). Positive and negative controls were run concurrently. The immunohistochemical reactions/expressions were assessed as weak or strong by visual inspection in comparison with positive and negative controls. All procedures were conducted in a fully accredited clinical laboratory with all relevant positive and negative controls. None of the markers studied are present in appreciable concentrations in normal lung tissue (Figure 2G).

## 5. Conclusions

We previously reported that the early lesion of post-primary TB is a distinct disease entity [11]. The present studies further suggest that early lesion is the bacteria’s offense, while granulomas are the host’s defense. Granulomas are widely considered to be the hallmark of TB. However, the finding of secreted mycobacterial antigens in multiple cells of the early lesions together with markers of multiple immune regulatory pathways suggest that these subclinical asymptomatic lesions are actually the major drivers of clinical PPTB. They accumulate both sensitized T cells and mycobacterial antigens in preparation for necrotizing hypersensitivity reactions. In multiple studies, effective vaccines and host directed therapies caused lessening of the PPTB-like pathology [18,20,25,29,32,33,34].

The early lesions produce cavities from which the organisms can escape to find new hosts, while granulomas protect the host from disseminated infection. As an obligate human parasite, *M. tuberculosis* needs both to survive. It must simultaneously protect its host and develop means to escape to new hosts. If protection by granulomas fails, the host dies of disseminated infection and the organisms die with it. *M. tuberculosis* only survives if post-primary lesions induce a cavity from which it can escape to find new hosts and the host remains healthy enough to circulate in the community.

The study of post-primary lesions will remain challenging because of the paucity of informative human tissues that can be used both to study the disease and to validate animal models. Fortunately, new multiplex technologies can measure proteins and nucleic acids on slides with a depth and precision undreamed of a decade ago [35]. Such technologies are needed to finally study these critical lesions of tuberculosis and to develop more effective interventions.

## Figures and Tables

**Figure 1 pathogens-10-01572-f001:**
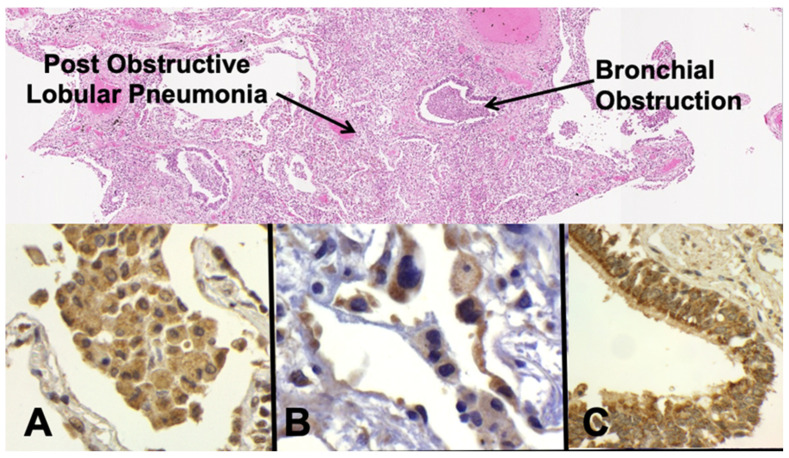
MTB Antigen in the Early Lesion. PPTB begins as an alveolitis that spreads subclinically as an obstructive lobular pneumonia for 1–2 years before undergoing necrosis as caseous pneumonia to initiate clinical PPTB. Evidence suggests that MTB uses its secreted antigens to direct these lesions towards caseation and cavitation that can transmit infection to new hosts (H&E stain 20×, partly reproduced from [6]). (**A**) Mycobacterial secreted antigens in alveolar macrophages (Immunostain 600×). (**B**) Mycobacterial antigens alveolar lining cells types 1 and 2 (Immunostain 600×). (**C**) Mycobacterial antigens in ciliated bronchial cells (Immunostain 400×).

**Figure 2 pathogens-10-01572-f002:**
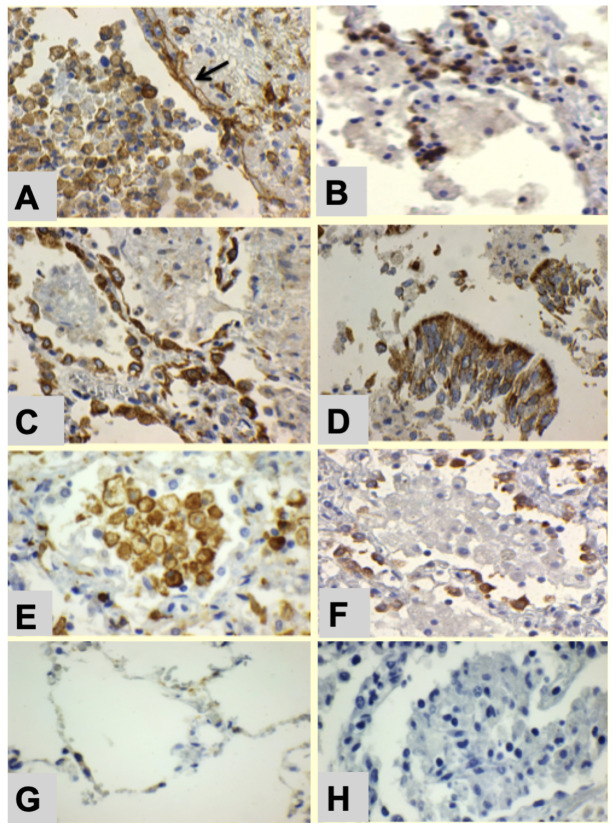
Characterization of the Early Lesion of PPTB. Immunostains for PD-L1, PD-1, COX-2, CD163, and FAS provide evidence that MTB use its secreted antigens to manipulate the host’s responses in the early lesion. (**A**) PD-L1 expression on multiple cell types including alveolar pneumocytes (ARROW) and alveolar macrophages (600×). (**B**) PD-1 expression on lymphocytes in adjacent alveolar walls (600×). (**C**) COX-2 in reactive alveolar pneumocytes (600×). (**D**) COX-2 in bronchiolar epithelium (600×). (**E**) CD163 staining of alveolar macrophages marking them as M2 cells (400×). (**F**) FAS expression in the reactive alveolar pneumocytes surrounding the alveolar macrophages (400×). (**G**) Representative adjacent nearly normal alveoli that shows little or no staining of any of the markers (600×). (**H**) Negative control on a lung with early PPTB shows no staining (600×).

## Data Availability

The data are not publicly available because they are histologic slides and patient data.

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
