# Peer review of "Early Lesion of Post-Primary Tuberculosis: Subclinical Driver of Disease and Target for Vaccines and Host-Directed Therapies"

_pathogens, 2021, doi:10.3390/pathogens10121572_

Round 1

Reviewer 1 Report

I don't have any comment.

Author Response

please check the attached file, thank you for your valuable comments

Reviewer 2 Report

Reviewer comments

Early lesion of post-primary tuberculosis: subclinical driver of disease and target for vaccines and host-directed therapies

Robert E. Brown and Robert L. Hunter

Comments

Overall commentary
Identification of host-directed therapies and vaccine targets to control Mycobacterium tuberculosis (Mtb) related pathologies are priority research areas that will greatly contribute to the global efforts to end tuberculosis disease. While there are swathes of literature on host immune responses to Mtb infection, this data is largely generated from studies of the host circulatory system, often measuring immune readouts that are distal from the site of infection, the lung. Experimental animal models of Mtb infection have been useful in understanding the responses at the site of disease. However, the trajectory of granuloma formation in these animals do not fully mimic the development of these bacteria containment structures as observed in humans.

Brown and Hunter have made observations in human lung tissue that are not possible in animal models because of the differences in trajectory of granuloma development between humans and the experimental animal models. The authors make observations that lead them to conclude that Mtb uses its secreted antigens to coordinate prolonged subclinical  development of the early lesions in preparation for necrotising reaction sufficient to produce a cavity, post-primary granulomas and fibrocaseous disease. This conclusion is agreeable as macrophages can take up Mtb antigens and this initiates a cascade of immune cell infiltration resulting in the formation of granulomas.

Specific comments:

  1. If this mechanism described leads to transmission of TB when the post-primary granulomas are coughed out, at what stage do intact bacteria become part of the post-primary granuloma so that when coughed out transmission will occur. In other words, does post-primary granuloma structure allow movement materials inside e.g bacteria and other immune cells or the other elements of the structure are only added externally.

  1. What is the source of these Mtb antigens, are the antigens secreted by live bacteria or they come from dying bacteria? If they are coming from dead bacteria, would this mean that TB treatment would have a role in creating these post-primary granulomas?

  1. Appropriate controls needed for the figures. While the authors show Mtb antigens and immune cells/markers, these images luck accompanying controls. For example, the reader has no idea how cells that do not express markers measured differ from cells shown. The authors should stain pieces of the same lung that do not have post-primary granulomas and put side by side for comparisons. In line 192, the authors indicate that positive and negative controls were run concurrently but these control images are not shown. In addition, the figure should be labelled with arrows and/or arrowheads especially for stains that are not very distinct for example figure 2A.

Minor comments

Line 13: change under to undergo

Line 19: change use to uses

Line 64: change ‘as far’ to ‘as well as’

Line 104: change ‘insure’ to ‘ensure’

Line 112: Delete ‘fully developed’

Line 173: change ‘ASO/M721E’ to ‘M72/AS01E’

Line 184: change ‘Consequentlhy’ to ‘Consequently’

Author Response

(The authors gave the same response as above.)

Reviewer 3 Report

The relevance of this study is well stated. Studying the early lesions of Post-Primary tuberculosis will be key in identifying new targets. 

I do have one question about Mtb: Have the Mtb strains been identified or characterized for the 25 samples? Could any of them be resistant strains? 

You do state that the samples have been previously described. However, I could not find much information about the Mtb. 

Author Response

Please check the attached file, thank you for your valuable comments.

Round 2

Reviewer 2 Report

Refer to my initial review comments which are NOT substantially addressed. 

Author Response

Responses are bold and underlined

Each of the Open Review  comments has bee addressed in the text. It is now improved.

Comments and Suggestions for Authors

Overall commentary
Identification of host-directed therapies and vaccine targets to control Mycobacterium tuberculosis (Mtb) related pathologies are priority research areas that will greatly contribute to the global efforts to end tuberculosis disease. While there are swathes of literature on host immune responses to Mtb infection, this data is largely generated from studies of the host circulatory system, often measuring immune readouts that are distal from the site of infection, the lung. Experimental animal models of Mtb infection have been useful in understanding the responses at the site of disease. However, the trajectory of granuloma formation in these animals do not fully mimic the development of these bacteria containment structures as observed in humans.

Brown and Hunter have made observations in human lung tissue that are not possible in animal models because of the differences in trajectory of granuloma development between humans and the experimental animal models. The authors make observations that lead them to conclude that Mtb uses its secreted antigens to coordinate prolonged subclinical  development of the early lesions in preparation for necrotising reaction sufficient to produce a cavity, post-primary granulomas and fibrocaseous disease. This conclusion is agreeable as macrophages can take up Mtb antigens and this initiates a cascade of immune cell infiltration resulting in the formation of granulomas.

Specific comments:

  1. If this mechanism described leads to transmission of TB when the post-primary granulomas are coughed out, at what stage do intact bacteria become part of the post-primary granuloma so that when coughed out transmission will occur. In other words, does post-primary granuloma structure allow movement materials inside e.g bacteria and other immune cells or the other elements of the structure are only added externally.

This illustrates a common misconception. Cavities form by dissolution of caseous pneumonia, not by erosion of granulomas. Bacteria grow in masses on the wall of the cavity where they can be coughed out to cause transmission of infection, This is more clearly explained in the first paragraph and also in the discussion of the revised manuscript.

  1. What is the source of these Mtb antigens, are the antigens secreted by live bacteria or they come from dying bacteria? If they are coming from dead bacteria, would this mean that TB treatment would have a role in creating these post-primary granulomas?

The antigens are secreted by live bacteria.   Furthermore, secreted antigens are requited for the virulence of M. tuberculosis.  This is now noted in the text.

  1. Appropriate controls needed for the figures. While the authors show Mtb antigens and immune cells/markers, these images luck accompanying controls. For example, the reader has no idea how cells that do not express markers measured differ from cells shown. The authors should stain pieces of the same lung that do not have post-primary granulomas and put side by side for comparisons. In line 192, the authors indicate that positive and negative controls were run concurrently but these control images are not shown. In addition, the figure should be labelled with arrows and/or arrowheads especially for stains that are not very distinct for example figure 2A.

Control immunohistochemistry images and text have been added. An arrow has been added to Figure 2A as suggested.

Minor comments

Line 13: change under to undergo          DONE

Line 19: change use to uses                    DONE 

Line 64: change ‘as far’ to ‘as well as’   DONE

Line 104: change ‘insure’ to ‘ensure’      DONE

Line 112: Delete ‘fully developed’            DONE

Line 173: change ‘ASO/M721E’ to ‘M72/AS01E’               DONE

Line 184: change ‘Consequentlhy’ to ‘Consequently’     DONE